# FiPPiE: A Computationally Efficient Differentiable method for Estimating Fundamental Frequency From Spectrograms

*Lev Finkelstein, Chun-an Chan, Vincent Wan, Heiga Zen, Rob Clark*

Google

{finklev, cachan, vwan, rajclarck}@google.com

## Abstract

In this paper we present FiPPiE, a Filter-Inferred Pitch Posteriorgram Estimator – a method of estimating fundamental frequency from spectrograms, either linear or mel, by applying a special kind of filter in the spectral domain. Unlike other works in this field, we developed a procedure for training an optimized filter (or kernel) for this type of estimation. FiPPiE, based on this optimized filter, demonstrated itself as a reliable fundamental frequency estimator that is computationally efficient, differentiable, and easily implementable. We demonstrate the performance of the method both by the analysis of its behavior on human recordings, and by the stability analysis with help of an automated system.

**Index Terms**: text-to-speech, pitch tracking, frequency estimation, signal processing

## 1. Introduction

Modeling fundamental frequency ($F_0$) plays an important role in the text-to-speech (TTS) field, since it reflects the melodic pitch pattern at which speech is spoken. In some TTS models we would like to have $F_0$ a part of the loss function. However, if such a system is not directly designed to predict prosodic characteristics including $F_0$, then such $F_0$ estimators should be both differentiable and fast enough for training purposes.

We introduce FiPPiE, a Filter-inferred Pitch Posteriorgram Estimator – a method of estimating $F_0$ from spectrograms, either linear or mel, by applying a special kind of filter in the spectral domain. While other works on pitch extraction in the spectral domain like PEFAC [1] and SWIPE [2] use analytically derived filters, we developed a procedure to train an optimized filter based on a set of potentially imperfect frequency trackers. The resulting filter has a form that is different from the analytical models, and we have demonstrated that the optimized filter can be approximated as a combination of Gaussian peaks of different amplitudes. Note that we describe $F_0$ estimation only; prediction of the voiced-unvoiced decision is beyond the scope of this paper.

In this paper we describe the methodology for training the optimized filter, alternatives for modeling the best filter shape, and finally we present FiPPiE that is developed using this methodology. FiPPiE is computationally efficient, differentiable, can be easily implemented, and also importantly can be easily debugged. Also, similar to extractors using spectral methods, it may be applied when only spectral information is accessible, i.e., where no time-domain waveform is available.

FiPPiE's performance is evaluated using two different types of experimentation. First, we perform a series of evaluations on a proprietary corpus of human recordings. In addition we use the TUSK automated framework [3] to show FiPPiE's robustness to various types of speech and noise.

## 2. Related Work

There are multiple approaches available for estimation of $F_0$. Some of them are based on the analysis in the audio domain. Examples include YIN [4], its probabilistic modification PYIN [5], RAPT [6] and REAPER [7] that are widely used in the TTS field. However, there is no differentiable version of those, which makes them inapplicable for being used during the training process.

Some methods are using deep learning for detecting the fundamental frequency from audio signals, e.g., CREPE [8], which is a convolution neural network used for $F_0$ estimation. However, precise models contain a large number of parameters, which makes them less computationally efficient.

Our approach belongs to the methods of estimating the fundamental frequency in the power-spectrum domain. One of the advantages of these methods is not requiring audio signal. For example, Tacotron-based systems [9] produce an output in power-spectrum domains, which makes it impossible to use an audio-based $F_0$ estimator as a loss function. There are few approaches that are simultaneously efficient, differentiable and applicable on spectral-only input. Some other high-precision methods like, for example, Harvest [10], do not fit these requirements.

Our work is very close to PEFAC [1] that, in addition to other signal processing techniques, uses a filter with a specially chosen impulse response. Another close comparison is SWIPE [2, 11], which uses a similar approach based on a modified cosine mask. There are also methods that explore harmonics behavior, e.g., subharmonic summation [12] that uses an exponentially decreasing factor for harmonics weights, or summation of residual harmonics (SRH) [13] that is both lightweight and provides a voicing decision.

Similarly to PEFAC and SWIPE, FiPPiE seeks for the $F_0$ that optimizes a response to some filter applied to the spectrogram. Unlike these works, however, we do not assume any hypothesis on the filter's form, except that it should reflect the harmonics nature of the speech signal. Our setup contains two very important features: (1) we use weights that play the role of priors in the analysis, (2) we use an optimized filter and optimized weights. We also show that the optimized kernel has a different form than the theoretical kernels used in these methods.

## 3. Methodology

The fundamental frequency in speech can be observed in the periodic pattern in the frequency analysis along the frequency

axis. We formulate the $F_0$ extraction task as an optimization of the response to a convolution-like operator with the kernel reflecting the harmonic structure of the signal. We use minimal assumptions on how this kernel (or filter) should look. We start with the most general formulation, and show how the filter may be trained using real data.

### 3.1. Formal setup

Let $S$ be a linear spectrogram, $S_t(i)$ be the signal value on the frame level defined on the set of the spectrogram bins, $hz(i)$ be the value corresponding to the $i$-th bin in hertz, and $t$ be the frame number in the time domain. Let $h_1, h_2, h_3, \ldots, h_L$ be the fundamental frequency ($F_0$) hypotheses in hertz – for example, a linearly spaced set of hypotheses. We adopt a posterior approach, meaning that we are looking for a set of masks $g_h : \mathcal{R} \Rightarrow \mathcal{R}$, such that the dot product $S_t(i) \cdot g_h(hz(i))$ reaches its maximum for $h$ corresponding to the real $F_0$ value. Intuitively, extremum points of $g_h$ providing the best response should fit to the extremum points of $S_t$.

To model a harmonics signal, we assume the existence of a unit mask, or a kernel, $g_u : \mathcal{R} \Rightarrow \mathcal{R}$, such that every $g_h(x)$ shares the same form of $g_u(x)$ with stretched frequency axis[1], i.e. $g_h(x) = g_u(x/h)$. We assume nothing on the nature of $g_u(x)$, but since higher frequencies are often noisier, we set $g_u(x)$ to 0 starting at a certain point. As a rule of thumb, if we're interested in $M$ first harmonics, $g_u(x)$ should be 0 starting $x = M + 0.5$.

Response $Z(h)$ per hypothesis $h$ can be represented as

$$Z(h) = \sum_i S_t(i) g_u(hz(i)/h), \tag{1}$$

where the summation is performed over the spectrogram bins. For the mel spectrogram case the problem formulation remains exactly the same, up to a different bin-to-frequency transformation of the spectrogram bins in Equation (1):

$$Z_{\text{mel}}(h) = \sum_i S_t(i) g_u(hz_{\text{mel}}(i)/h), \tag{2}$$

where $S_t(i)$ is the mel spectrogram value at bin $i$, and $hz_{\text{mel}}(i)$ is the value of the $i$-th mel bin in hertz.

Rather than looking for $F_0$ by directly maximizing the response, we take into account the fact that some of the hypotheses are more meaningful than others. For example, if the hypotheses of 50Hz and 100Hz have similar responses, we should prefer the one of 50Hz, since 100Hz is probably just its higher harmonics. In addition, the $F_0$ distribution has some priors in the real world, that we would also like to take into account. To reflect this preference, we introduce *hypotheses weights* $w(h)$, and our optimization criterion becomes[2]

$$F_0 = \arg\max_h [w(h) Z(h)] \tag{3}$$

Instead of deriving kernel $g_u(x)$ and weights $w(h)$ analytically, we train them on real data as shown in the following sections.

---

[1]Alternatively, we may switch to the log domain and use a convolution representation like in PEFAC, but the current formulation was a better fit for our model and usage mode.

[2]argmax function is not differentiable, so for differentiability, we use the standard argmax approximation using the softmax function.

### 3.2. Spectrogram setup and preprocessing

In our setup we targeted mel spectrograms with 128 mel bins with a 50ms spectrogram window. In addition, we (1) removed low-order DCT components (in our case, the 3 lowest components) to filter the corresponding noise, and (2) considered only the first 40 mel bins, since higher frequencies were noisy and had no significant impact on determining $F_0$.

### 3.3. Training the optimized kernel and weights

We used two proprietary read speech corpora in US English, $\mathcal{C}_1$ and $\mathcal{C}_2$, by professional voice actors[3]. $\mathcal{C}_1$ contained English speech, read in a natural style, by 58 speakers with different accents (mostly North American accent), while $\mathcal{C}_2$ contained the utterances of 40 speakers in North American accent only.

We randomly collected 1000 training and 1000 test utterances from $\mathcal{C}_1$. Only voiced frames were collected, resulting in approximately 180K frames for training and a similar number for testing. Since we didn't have groundtruth data for this type of training, we used a combination of a modified PYIN [5] and REAPER [7] trackers to set $F_0$ values. In our experiments we observed that PYIN was more reliable in low pitch voices, but was underperforming for high pitch voices, so we used PYIN's value for 200Hz or below, and REAPER's value for above 200Hz. REAPER was used for the voicing decision. In addition, we collected 1000 utterances from $\mathcal{C}_2$, to be used for validation on unseen speakers.

The training was performed as a regression task, while we validated various filter and weight setups. The fundamental frequency was defined by Equation 3 using the mel-based response described by Equation 2. We optimized the mean absolute error (MAE) loss, but other types of loss functions demonstrated similar behavior. The hypotheses $h_i$ were linearly spaced from 50Hz to 600Hz with the step of 5Hz, overall $L = 111$ hypotheses[4]. We considered a kernel $g_u(x)$ to be a vector of values from 0 to 5.5 with the step 0.01 (up to 5 harmonics, $N = 551$ of non-zero elements; linear interpolation used for the rest of the values). The weights $w(h)$ were a vector of $L$ elements, one per hypothesis $h_i$.

We considered different forms of constraints on the filter and the weights. We started with a **free-form kernel optimization** with no constraints at all. This type of a task requires $N = 551$ kernel and $L = 111$ weight parameters mentioned above. The results, shown in Figure 1, demonstrate a harmonic trend in the optimized kernel. Decreasing the weights is also intuitive – lower $F_0$ hypotheses are typically preferred over their harmonics. They also reflect the priors of the general $F_0$ distribution.

To explicitly emphasize the harmonic nature of the optimized kernel, we introduced a **piecewise-monotonic** constraint where the kernel function is monotonically non-decreasing in $[i - 0.5, i]$, and monotonically non-increasing in $[i, i + 0.5]$. The number of the parameters is the same, and the results are shown in Figure 2. It is possible to see that there is no negative component in this type of a kernel, and that was very consistent in all the similar experiments.

We also tested a special **piecewise-cosine kernel** that behaved as cosine functions $a_{2i} \cos(2\pi x) + b_{2i}$ in each half-period $[i - 0.5, i]$ and as $a_{2i+1} \cos(2\pi x) + b_{2i+1}$ in $[i, i + 0.5]$, with

---

[3]The corpora meet the Google AI Principles https://ai.google/principles/.

[4]A further increase of the resolution to 1Hz did not have any significant impact on the performance.

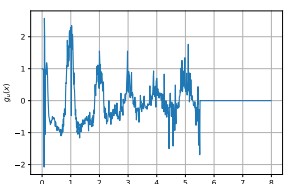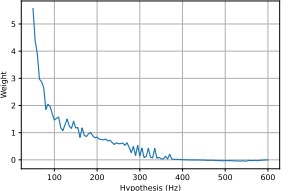

Figure 1: *Optimized kernel (left) and optimized weights (right) for the free-form optimization*

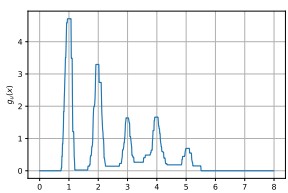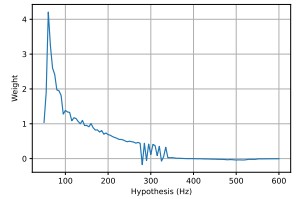

Figure 2: *Optimized kernel (left) and optimized weights (right) for the piecewise-monotonic constraints*

amplitudes $a_k$ and bias $b_k$ differing between harmonics. We imposed constraints on these parameters to ensure continuity between the segments. The number of kernel parameters for 5 harmonics is 11 (including half-harmonics at 0), and the results are shown in Figure 3.

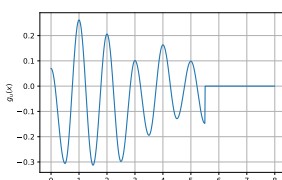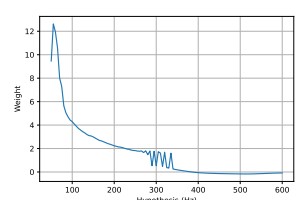

Figure 3: *Optimized kernel (left) and optimized weights (right) for the piecewise-cosine constraints*

Finally, driven by the results of the piecewise-monotonic optimization, we used a kernel that was a combination of Gaussian masks with the same deviation and different amplitudes, centered at $i$, and the results are shown in Figure 4. Only 7 parameters (one for deviation, and 6 for amplitudes) are required in this case.

The results of the loss of each one of the methods[5] are shown in Table 1. While the free-form optimization provided the best results, more general models prevent overfitting. We can see that the Gaussian mask version was very close to the piecewise-monotonic version, while the piecewise-cosine's performance was worse. This was an evidence to a (maybe non-intuitive) observation that the optimized kernel does not gain from the negative component of the filter, meaning that for human speech this component is probably less important.

### 3.4. Practical model for FiPPiE

The number of speakers in the training dataset was limited, so to prevent overtuning, we decided to reduce the number of parameters as much as possible and not to use the free-form kernel. On

---

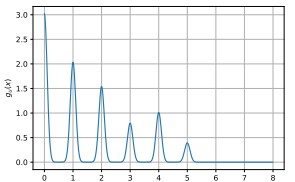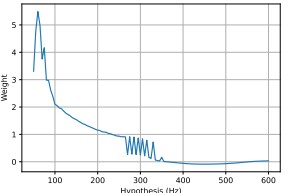

Figure 4: *Optimized kernel (left) and optimized weights (right) for the Gaussian masks constraints*

Table 1: *MAE loss for different types of constraints*

|  | Train set | Test set | Validation set |
| --- | --- | --- | --- |
| Free-form | 3.38 | 3.44 | 3.93 |
| Piecewise-monotonic | 3.69 | 3.68 | 4.24 |
| Piecewise-cosine | 4.11 | 4.07 | 4.69 |
| Gaussian mask | 3.85 | 3.77 | 4.35 |

the other hand, the second-best model (piecewise-monotonic) was only slightly better than the Gaussian mask kernel with a smaller number of parameters, so we decided to adopt the latter as the kernel model. However, it still used $L = 111$ parameters for the weight representation, which was too large to rule out a possibility of potential overtuning, especially given the lack of the representation for some pitch values.

In order to stay on the safe side, we decided to parameterize the weight curve as well. The weight shape resembled the lognormal distribution (see Figure 4), which requires only 4 parameters, that we trained together with the kernel. Such a reparameterization led to some performance degradation, but the very small amount of parameters (7 parameters for kernel, and 4 parameters for weights) justified that change. Further work of finding a better fit for the hypothesis weights is in process.

## 4. Experimentation

To demonstrate the performance of FiPPiE, we conducted two sets of experiments. The first set performed a qualitative analysis of $F_0$ tracking on the proprietary corpora $\mathcal{C}_1$ and $\mathcal{C}_2$. The second set used the TUSK framework [3] to automatically compare the characteristics of $F_0$ trackers.

FiPPiE was applied to mel spectrograms. While the training was done on spectrogram windows of 50ms, the best performance during the experimentation was obtained by using the window size of 100ms, due to a higher robustness to low-pitch voices. There was no significant impact on retraining the kernel on 100ms spectrograms. See more in Section 5.

### 4.1. Comparative performance

In this set of experiments we compared the performance of different algorithms on $\mathcal{C}_1$ and $\mathcal{C}_2$. These corpora contain a large variety of voices collected under different conditions, so this type of comparison is very informative for evaluating a system that is (presumably) optimized for real data. Note that the speakers of $\mathcal{C}_1$ were used in the training, while the speakers from $\mathcal{C}_2$ were not.

We randomly selected 1000 utterances from each one of the corpora, and compared the performance of PYIN, REAPER, PEFAC, SWIPE, SRH, and FiPPiE. Since FiPPiE doesn't have its own voice decision algorithm, we decided to use REAPER's voicing decision verdict across all the algorithms since in our

experiments it was the most stable. We considered Gross Pitch Error (GPE) to be a mismatch measurement. GPE is the percentage of voiced frames with the relative mismatch is 20% or more, and it is common in pitch estimation (see, for example, [14], [15]).

Since we don't have a groundtruth $F_0$, we compared the estimated $F_0$ versus a consensus (median) of all 6 algorithms. Besides, since the median is not the real groundtruth, we provide two comparisons against PYIN and SWIPE respectively to better understand the behavior of different algorithms. The results are shown in Table 2. It is possible to see that the first

Table 2: *Gross Pitch Error for different pitch trackers on different corpora vs. different reference algorithms.*

| Corpus Reference | $C_1$ median | $C_2$ median | $C_1+C_2$ median | $C_1+C_2$ PYIN | $C_1+C_2$ SWIPE |
|---|---|---|---|---|---|
| FiPPiE | 4.5% | 1.4% | 3.0% | 2.1% | 5.4% |
| PYIN | 5.0% | 1.8% | 3.4% | - | 5.7% |
| REAPER | 6.0% | 4.0% | 5.0% | 3.3% | 6.7% |
| SWIPE | 5.8% | 1.6% | 3.7% | 5.7% | - |
| PEFAC | 7.8% | 4.0% | 5.9% | 8.1% | 4.9% |
| SRH | 7.9% | 4.0% | 5.9% | 8.2% | 5.0% |

corpus has a high level of mismatches. The reason is that this corpus is very diverse, the number of utterances is not balanced per speaker, plus different algorithms have different strengths and weaknesses. As the result, the median may not always be a reliable baseline. To make the results more focused, we picked 5 US English speakers with different characteristics from $C_1$, 8 speakers from $C_2$, randomly sampled 100 utterances for each, and repeated the experiment. The results are shown in Table 3

Table 3: *Gross Pitch Error for different pitch trackers on different corpora vs. different reference algorithms for the limited set of speakers.*

| Corpus Reference | $C_1$ median | $C_2$ median | $C_1+C_2$ median | $C_1+C_2$ PYIN | $C_1+C_2$ SWIPE |
|---|---|---|---|---|---|
| FiPPiE | 1.4% | 1.1% | 1.2% | 2.0% | 1.76% |
| PYIN | 1.7% | 1.3% | 1.5% | - | 2.02% |
| REAPER | 3.7% | 3.2% | 3.4% | 3.1% | 3.45% |
| SWIPE | 1.6% | 1.2% | 1.4% | 2.0% | - |
| PEFAC | 10.3% | 2.8% | 6.3% | 7.1% | 6.74% |
| SRH | 10.6% | 2.9% | 6.5% | 7.3% | 6.91% |

From the two experiments we can observe:

- The performance of FiPPiE vs. the median on these datasets is slightly better than the performance of PYIN and of SWIPE, while REAPER, SRH and PEFAC have a higher error.

- While the speakers from $C_1$ have been seen during training, FiPPiE's results on the two corpora are close enough for the second test (note that all the algorithms demonstrated better performance on $C_2$).

- Despite a similar error rate vs. the median, the estimations provided by FiPPiE, PYIN, and SWIPE are suggesting disagreement between them.

- PEFAC and SRH had a high GPE rate on $C_1$, especially in the second experiment. In-depth analysis showed that the reason was a vulnerability to a certain kind of low-pitch speech.

Overall, the experiments show that performance of FiPPiE on this dataset is at least as good as the performance of PYIN and SWIPE, and is better than that of REAPER, PEFAC, and SRH. Note, however, that this comparison is imperfect since we lack the groundtruth, and since we compare the performance only on the voiced frames marked by REAPER.

### 4.2. Automated performance analysis

In this section, we used the TUSK framework [3] for the analysis of FiPPiE behavior and its comparison with other $F_0$ estimators. TUSK was designed to analyze the behavior of $F_0$ trackers for different types of the target speech, thus removing the dependency on the differences in the datasets used for the comparison. TUSK considered six major parameters—temporal fluctuations, varying amplitudes and phases of harmonic components, white and pink noise, and reverberation—and analyzed the tracker behavior by modifying these parameters in some artificial signal. Due to the lack of varied groundtruth in $F_0$ tracking, using an automated framework permits comparison between different systems in a more objective way.

In this experiment, we compared FiPPiE with two other spectrogram-based methods – PEFAC and SWIPE. The original TUSK analysis is based on the "basic" fundamental frequency of 440Hz that is not typical for speech, so instead, we performed the analysis for the basic frequencies of 100Hz, 200Hz, and 300Hz. Due to the lack of space, we show graphs for only some of the findings.

Figure 5 shows the reconstruction abilities of the three methods for different frequencies on the clean signal[6]. We can

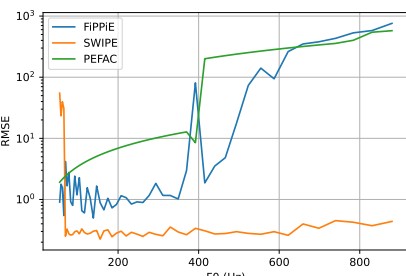

Figure 5: *General performance estimation*

see that SWIPE's performance on a clean signal is the best except for very low frequencies, while FiPPiE outperforms PEFAC.

Figure 6 shows the influence of temporal fluctuation in the $F_0$ contour (vibrato intensity) for the basic frequency of 200Hz. While SWIPE slightly outperforms FiPPiE at the beginning, its performance degrades rapidly. FiPPiE's behavior is better than PEFAC's as well. This trend is even stronger for the basic frequency of 100Hz, but for the high-frequency signal of 300Hz, FiPPiE's performance is also degraded by this parameter. This vulnerability of FiPPiE for higher frequencies is observed in other experiments as well, so it is possible that this is an inherent feature. We discuss it more in Section 5.

In Figure 7 we can see the impact of amplitude and phase. FiPPiE becomes less stable when the amplitude and phase start to increase, which is consistent with the idea of the kernel op-

---

[6]In the TUSK framework, the clean signal contour is built using the basic frequency with some additional fluctuation based on the model Klatt [16].

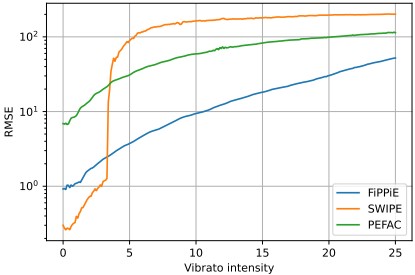

Figure 6: *Influence of vibrato intensity for $F_0 = 200Hz$*

timization – when the harmonics become "unnatural", FiPPiE fails. As before, its relative behavior improves for the low-frequency signal of 100Hz, but worsens for the high-frequency signal of 300Hz.

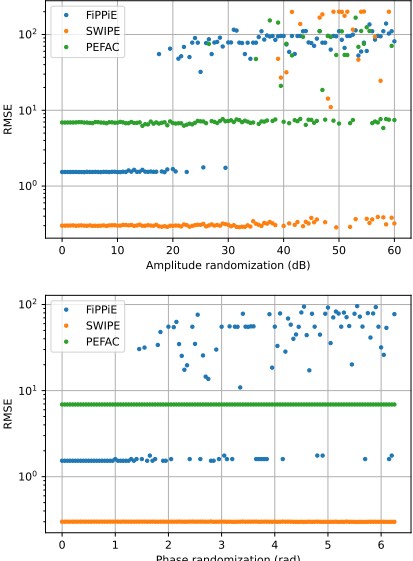

Figure 7: *Influence of the amplitude randomization (top) and of the phase randomization (bottom) for $F_0 = 200Hz$.*

Figure 8 show the robustness of the systems to white and pink noise. PEFAC is noise-robust, while SWIPE requires a pretty clean signal, especially for the low-frequency base signal. FiPPiE demonstrates good robustness for 100Hz and 200Hz basic frequency, but its behavior becomes closer to SWIPE's when this basic frequency is increased.

Finally, the vulnerability to reverberation is shown in Figure 9. In this experiment, FiPPiE's performance is worse than for the two other algorithms. For the lower basic frequency, this weakness disappears.

Overall, we can see that FiPPiE demonstrated a solid performance for the frequencies typical for the average human speaker. It outperformed PEFAC in the general evaluation, vibrato intensity experiment, and the low-level amplitude and phase randomization, and even had a close robustness to white and pink noise (PEFAC is considered very robust). If compared to SWIPE, FiPPiE was not as precise for the high-pitch signal, but behaved much better in the presence of noise, and was more stable in its response to the vibrato intensity parameter. FiP-

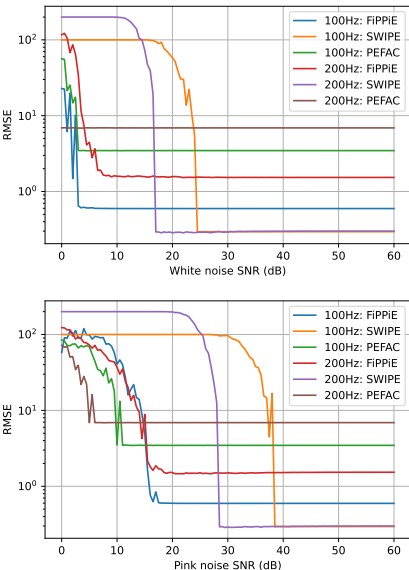

Figure 8: *Impact of the white noise (top) and of the pink noise (bottom) for $F_0 = 100Hz$ and $F_0 = 200Hz$.*

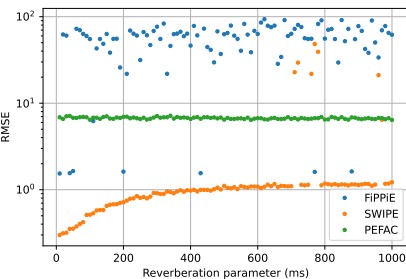

Figure 9: *Impact of the reverberation for $F_0 = 200Hz$*

PiE's major weakness is apparently robustness to reverberation and some vulnerability when the basic frequency is high.

One of the important advantages of FiPPiE is its ability to work reliably in low-frequency voices. It is also interesting to mention that FiPPiE's precision is not affected by the fact that it operates on the mel spectrogram level, that already assumes some precision loss.

## 5. Discussion

In this paper we presented a computationally efficient differentiable method for estimating fundamental frequency from spectrograms. During the development, we observed a few very challenging questions that we present in this section.

- **Linear vs. mel spectrograms:** Our method is oriented towards mel spectrograms. The motivation was its usage in Tacotron-like systems [9] that use mel spectrograms as an intermediate representation. The question rises, whether we could get more if we used linear spectrograms instead. When we tried to tune the method to process linear spectrograms, the topological shape of the optimal kernel and the weights remained almost the same. We were able to apply the same kernel and weights to the linear spectrograms without actual performance loss, presumably because of the similarity of

mel and linear spectrograms in the range of interest.

- **Low-pitch voices:** Low-pitch voices often have a somewhat unclear spectrogram structure in lower bins, which makes it difficult to get a good resolution. However, choosing a long window length (100ms in our case) helped to overcome this problem. A similar observation was done in other works as well – PEFAC paper [1] used a 90ms window, and SRH paper [13] used a 100ms window. It is interesting though that there was almost no difference between the kernels trained for 50ms and 100ms windows, so we didn't need to retrain the kernel. There are, however, specific low-pitch speakers that may cause FiPPiE to have isolated glitches.

- **High-pitch voices:** High-pitch voices of about 300Hz and higher are less stable for FiPPiE. We believe this may be caused by the lack of the training data in these voice ranges. This is also supported by the alternating weight function for optimized weights, see Figures 1, 2, 4 in the area near 300Hz.

- **Voices without a clear harmonic structure.** Some of the voices, like creaky voices (see different variations of them in [17]), often lack a clear harmonic structure and are therefore represent an inherent challenge for this family of methods. FiPPiE was able to handle many of such examples successfully, but glitches happened occasionally.

- **Other languages:** We did a limited evaluation with pitch tracking on other languages, however, we were not able to establish a stable reference since the experimentation suffered from two major issues: first, the voiced-unvoiced decision in these corpora turned out to be less reliable than that for US English, thus leading to incorporating unvoiced segments into the statistics, and second, there was too great a mismatch between the pitch trackers, so the median could not be used as a stable reference. Evaluating FiPPiE performance in other languages is ongoing research.

- **Voiced-unvoiced decision:** The decision on voiced vs. unvoiced segments is very important for the pitch tracker. However, the current research is devoted to a different aspect of analysis, namely to finding $F_0$ in voiced frames, so the voiced-unvoiced decision is beyond the scope of this paper.

## 6. Conclusions

In this paper we presented FiPPiE, a Filter-inferred Pitch Posteriorgram Estimator, that is capable of estimating the fundamental frequency in spectrograms, either linear or mel. FiPPiE is based on training the optimized filter and the optimized weights; we describe the training procedure and show that the optimized kernel, trained on real speech samples, is well approximated by a combination of Gaussian masks, which differs from the existing models that use filters that exploit the negative component as well (e.g., cosine-like filters).

FiPPiE demonstrated strong performance versus other $F_0$ trackers, despite that in our experiments it was applied to mel spectrograms. Experiments on the real database and experiments using an automated system showed FiPPiE's ability to be a reliable $F_0$ estimator in a variety of voices and conditions.

FiPPiE is computationally efficient, differentiable, and easily implemented, which makes it a good candidate to be a part of a loss function. It even can be used as a light-weight $F_0$ tracker, instead of potentially more precise, but either non-differentiable, or too computationally heavy methods.

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
