# OpenReview forum: "FiPPiE: A Computationally Efficient Differentiable method for Estimating Fundamental Frequency From Spectrograms"
_Interspeech.org/2023/Workshop/SSW — SSW12_

### Official Review · Reviewer_9TKL · 2023-05-30
**Decent paper on F0 estimation, could benefit from a more clear description of the basic principle of the method in the beginning of the paper**

**Rating:** 6
**Confidence:** 4

**Review:**

This paper describes a fundamental frequency (F0) estimation algorithm that is based on summation of the magnitude spectrum using learned filters and weights. The benefits of the proposed method are that it functions solely on the spectrogram domain, it is differentiable, and it has low computational complexity.

The description of the method in the paper is not very clear at first, especially if the reader is not familiar with various F0 estimation methods and their functioning. However, the method seems sound, although some descriptions are not clear or are missing. The authors conduct (limited) experiments to compare the proposed method to other F0 estimation methods. The results on accuracy show good performance, however, the proposed method does not seem to be particularly robust to various conditions, as is shows in the experiments.

The paper is of decent quality, but could benefit from better structure and clarity in writing in some parts. The work is similar to other works in the field, but has some originality. Overall, the paper is of sufficient interests to the public.

Detailed comments:

The idea of how the proposed method really works is not very clearly described in the beginning of the paper, due to which there might be a lot of confusion to the readers that are not that familiar with fundamental frequency estimation. It would make the paper better if the basic functioning principle of the proposed method was explained more clearly in the beginning.

Sec. 3.3: It’s not clear if the corpora C1 and C2 are single or multi-speaker datasets. If they were single speaker datasets, the generalizability of the method to other speakers is questionable. Later on in the paper it is explained that these are multi-speaker datasets, however, only very few speakers are selected. A more diverse dataset for training and evaluation would be beneficial.

Sec. 3.4: “Given the empirical results from the experiments above, we decided to use a Gaussian kernel with the weights parameterized with a lognormal distribution” It’s not clear how the previous experiments, with free-form mask giving the best results, lead to using a log-normal distribution. Please explain more.

Sec. 4.2: “Overall, we can see that FiPPiE demonstrated a very strong behavior for the frequencies typical for the average human speaker.” Seems like FiPPiE is often inferior in comparison to SWIPE and PEFAC in the automated performance analysis. Why do the authors think that FiPPiE “demonstrates a strong behavior”?

Sec 5: Linear vs. mel spectrograms: This discussion is a bit confusing since all the figures are depicting a linear spectrogram. It is not shown in the paper about how does the algorithm is formulated for Mel-spectrograms?

Sec. 5: “but glitches happened periodically.” Maybe a slightly unfortunate choice of wording for a paper that deals with periodicity :)

---

> ### Author Response · Authors · 2023-06-28
> **Thank you very much for your review!**
>
> This is the summary of what we did to address your comments:
>
> > It would make the paper better if the basic functioning principle of the proposed method was explained more clearly in the beginning.
>
> The reason was actually the lack of space (which is now not a problem due to the additional page). We elaborated an intuitive idea of the method at the beginning of Section 3.
>
> >  It’s not clear if the corpora C1 and C2 are single or multi-speaker datasets. If they were single speaker datasets… A more diverse dataset for training and evaluation would be beneficial.
>
> They are multi-speaker corpora; we added a description in section 3.3.
>
> > It’s not clear how the previous experiments, with free-form mask giving the best results, lead to using a log-normal distribution.
>
> This is a very good point. Having a very large number of parameters for the limited number of speakers in the training set was still a bit risky overfitting-wise. So, we decided  to sacrifice some additional performance for greneralization. We added clarification to Section 3.4.
>
> > Why do the authors think that FiPPiE “demonstrates a strong behavior”?
>
> You’re right, “very strong” is a bit too strong here. We replaced it by “solid performance” and rephrased the paragraph to justify the claim.
>
> > It is not shown in the paper about how does the algorithm is formulated for Mel-spectrograms?
>
> We clarified it (added Equation 2 explicitly). The figures actually correspond to the mel spectrogram training (mentioned at the beginning of 3.2), just they look very similar between the linear and the mel spectrograms (the difference between the two lies in their bin frequencies, but we translate everything to hertz, see Equations (1) and (2)). We added an explicit mention of the optimization criteria in Section 3.3.
>
> > “but glitches happened periodically.” Maybe a slightly unfortunate choice of wording for a paper that deals with periodicity :)
>
> Nice :) Rephrased.

---

### Official Review · Reviewer_aQ9B · 2023-06-06
**This is a new and interesting approach to a classical task. The performance is promising. The limitation is that it was tested on 13 US English speakers only. I propose the acceptance of the paper.**

**Rating:** 8
**Confidence:** 4

**Review:**

F0 tracking is a classical problem in speech processing. The proposed method works in the spectral domain that is advantageous for neural optimization tasks. The optimized filter is trained on a proprietary database. Unfortunately neither the training nor the validation database is publicly available. Also, even the basic parameters of the databases are missing (no. of speakers, gender, age, length. etc.).
Taking into account the 180k frames and 50ms frame length it seems that the training was performed on about 2,5 hours of US English voiced frames.
The performance comparisons on both the human recordings and the TUSK framework are convincing and well described.
It is a pity that the extreme conditions (lowest pitch range, e.g. below 80Hz or the highest e.g. above 350Hz, mixed excitation) are not analyzed in detail (except Fig. 5).
Availability of at least part of the infrastructure (code, filer/kernel) would help in reconstructing the results and further improvement of the approach.
Nevertheless I find the paper interesting and propose to accept it.

---

> ### Author Response · Authors · 2023-06-28
> **Thank you very much for your review!**
>
> This is the summary of what we did to address your comments:
> > The limitation is that it was tested on 13 US English speakers only.
>
> The reason was that the corpus was too diversed, and the median should be taken with the grain of salt. However, it is a very good point, so we added this experiment as well.
>
>  > The basic parameters of the databases are missing (no. of speakers, gender, age, length. etc.).
>
> We added corpus descriptions.
> > It is a pity that the extreme conditions (lowest pitch range, e.g. below 80Hz or the highest e.g. above 350Hz, mixed excitation) are not analyzed in detail (except Fig. 5).
>
> This is a very good point. Unfortunately, we don’t have a reliable baseline since the median for extremely low and extremely high pitch voices was very flaky (the algorithm have different strengths and weaknesses). We tried to do this type of slicing in our experiments, but the data was too unstable to include it into the paper, so we decided to rely on TUSK.
>
> > Availability of at least part of the infrastructure (code, filer/kernel) would help in reconstructing the results and further improvement of the approach.
>
> We’re checking the options to make it an open source.

---

### Decision · Program_Chairs · 2023-06-14

**Decision:**

Accept

**Comment:**

SSW2003 received 45 papers. The acceptance rate is 82%. We are pleased to inform you that your paper has been accepted by the SSW2023 Program Committee. Please read the reviews carefully and submit your camera-ready paper by June 28th. Most reviewers performed a detailed review. Please answer to their questions and consider their comments. Note that camera-ready papers are credited with one extra page to allow authors to consider reviewers’ suggestions. So max 7 pages in total including figures & refs.
The deadline for submitting the revised version (with full non-anonymized authors and refs!) is 28th June.